# Venous Thromboembolism following Two Doses of COVID-19 mRNA Vaccines in the US Population, 2020–2022

**DOI:** 10.3390/vaccines10081317

**Published:** 2022-08-15

**Authors:** Daoyuan Lai, Yan Dora Zhang, Junfeng Lu

**Affiliations:** 1Department of Statistics and Actuarial Science, Faculty of Science, The University of Hong Kong, Hong Kong 999077, China; 2Centre for PanorOmic Sciences, Li Ka Shing Faculty of Medicine, The University of Hong Kong, Hong Kong 999077, China; 3First Department of Liver Disease, Beijing You’An Hospital, Capital Medical University, Beijing 100069, China

**Keywords:** adverse drug reaction, COVID-19 mRNA vaccines, venous thromboembolism, VAERS

## Abstract

The COVID-19 mRNA vaccine is one of the most effective strategies used to fight against COVID-19. Recently, venous thromboembolism (VTE) events after COVID-19 mRNA vaccination have been reported in various research. Such a concern may hamper the ongoing COVID-19 vaccination campaign. Based on the US Vaccine Adverse Event Reporting System data, this modified self-controlled case series study investigated the association of COVID-19 mRNA vaccination with VTE events among US adults. We found the VTE incidence rate in the recommended dose interval does not change significantly after receiving COVID-19 mRNA vaccines. This conclusion still holds if the analysis is stratified by age and gender. The VTE onset may not be significantly associated with COVID-19 mRNA vaccination.

## 1. Introduction

The COVID-19 pandemic has lasted over two years and has caused massive devastation. COVID-19 vaccines, such as BNT162b2 and mRNA-1273, provide hope for ending this pandemic. Some research has linked various adverse events (AEs), such as venous thromboembolism (VTE), to COVID-19 mRNA vaccination [1,2,3,4]. This safety signal, however, is debatable because recent cohort research found that VTE is not substantially related to COVID-19 immunization [5]. Clarifying the uncertainty around this safety signal is beneficial in overcoming the general public’s aversion to vaccination.

Because clinical trials have insufficient power to detect the signal of many rare AEs, large-scale post-marketing surveillance is still essential in evaluating vaccination safety. The Vaccination Adverse Event Reporting System (VAERS) is a nationwide passive surveillance system that analyzes potential vaccine safety issues in the United States [6]. The VAERS gathers AE reports voluntarily submitted by the general population in the United States. Because of the broad and diversified population covered, the VAERS can quickly uncover potential safety risks. On the other hand, because the VAERS is a publicly available dataset, it may garner more public trust. However, research on the VAERS dataset is frequently hampered by data limitations, such as case underreporting and the lack of a control group to assess excess vs. baseline risk in the population of interest [7]. A report from the US Institute of Medicine found insufficient evidence to accept or deny a causal association for 85 percent of vaccine-adverse event pairs detected in the VAERS database [8].

This paper investigated the association between VTE and COVID-19 mRNA vaccination in the US population by performing self-controlled case series (SCCS) analysis on the VAERS data [9]. The SCCS is a popular design used for estimating the risk of adverse events following COVID-19 vaccination [3,10,11,12]. The SCCS method compares event incidence rates when a person is exposed (i.e., risk period) to event incidence rates when the same person is not exposed (i.e., reference period). Because of this design, the SCCS method does not need a control group, and comparisons are made within the case series. However, the original SCCS method cannot be directly applied to VAERS data as the event is dependent on the exposure (i.e., underreporting). To address this problem, a modified SCCS was proposed with rigorous justification [13,14,15]. This modified SCCS method has been applied to passive surveillance systems to evaluate the risk of intussusception following a monovalent RV1 (Rotarix, GlaxoSmithKline Biologicals) vaccine or a pentavalent RV5 (RotaTeq, Sanofi-Pasteur MSD) vaccine [13,14,15]. We discussed whether the risk of developing VTE is significantly elevated following the two doses of COVID-19 mRNA vaccination compared to the reference period.

## 2. Materials and Methods

### 2.1. Study Design and Data Sources

The VAERS dataset was used for our investigation. The VAERS is a national pharmacovigilance database co-managed by the US Centers for Disease Control and Prevention and the US Food and Drug Administration since 1990. Anyone, including healthcare professionals, vaccine makers, patients, and others, can submit an AE report to the VAERS.

VAERS data are subjected to some limitations, including underreporting and the failure to determine a control group. We performed a modified SCCS analysis [9] in the VAERS dataset to solve this problem. The SCCS approach, as the name implies, does not require a control group. The SCCS approach calculates the AE’s relative incidence (RI). If the RI is significantly larger than one, the risk of developing VTE in the risk period is significantly higher than in the reference period. Informed consent was waived as the VAERS dataset was de-identified. Thus, no personally identifiable information was available.

### 2.2. Procedures

As the individual-level VAERS reports are publicly available, we were able to extract a case series of VAERS reports of VTE related to COVID-19 mRNA vaccination. First, we extracted VTE cases related to COVID-19 mRNA vaccination in the VAERS database using the following preferred terms for VTE in the Medical Dictionary for Drug Regulatory Activities [16]: “Thrombosis”, “Deep vein thrombosis”, “Mesenteric vein thrombosis”, “Cerebral venous thrombosis”, “Cavernous sinus thrombosis”, “Intracranial venous sinus thrombosis”, “Pulmonary embolism”, “Embolism venous”, “Axillary vein thrombosis”, and “Venous thrombosis”.

For each VTE report, age, reported gender, kind of vaccine, vaccine dosages, immunization date, time to VTE onset, and seriousness were extracted. Serious cases were defined as significant disability/incapacity, hospitalized, under life-threatening conditions, and death. Only cases with onset times later than the immunization time were kept. Duplicate cases were eliminated. The time to VTE onset was noted, and our analysis was limited to those that developed VTE within the indicated dose interval (21 days for the BNT162b2 vaccine and 28 days for the mRNA-1273 vaccine).

Clinicians were involved in the medical review process. Cases were classified as presumptive VTE (including pulmonary thromboembolism) cases if they had clinical signs and symptoms (e.g., chest pain, dyspnea, tachycardia, pain and swelling in the limbs, difficulty walking, skin discoloration) plus diagnostic testing (e.g., vascular ultrasound or CT scan), diagnostic testing only, or a physician’s diagnosis or impression of VTE (including pulmonary thromboembolism). Reports in which clinicians reported that the patient did not have VTE were discarded. Cerebral thrombosis, artery thrombosis, myocardial infarction, and cordis mural thrombus were all excluded. A visualization of our data cleaning procedure is offered (Appendix A).

We could not use the Brighton Collaboration criteria to evaluate most VAERS reports, since the medical records were unavailable. Sensitivity analysis, which assumed that 80% of cases could ultimately satisfy the Brighton Collaboration criteria for VTE, was performed to compensate for this limitation [17].

### 2.3. SCCS Analysis

The modified SCCS analysis compared the VTE incidence rate in the risk period to the reference period. The observation period for each COVID-19 mRNA vaccine was the recommended dose interval (i.e., 0–21 days for the BNT162b2 vaccine and 0–28 days for the mRNA-1273 vaccine), with day 0 being the day of vaccination. There were four observation periods: 0, 1–7, 8–14, and 15–21 (or 28) days. The first three periods were risk periods, followed by the reference period (Appendix A). This risk period design is comparable to the one used in another statewide SCCS study in England [3].

The RIs, defined as the incidence rate of myocarditis in the risk period relative to the reference period for each dose, were estimated via two methods. The first parametric approach assumed that the probability of reporting an AE to VAERS drops exponentially with time and reports the RIs after each dose. An RI greater than one indicates that the risk period’s AE incidence was considerably higher than the reference period following each dosage. The second non-parametric approach calculates the ratio of RIs after the second and first doses. If the ratio of RIs was more than one for a specific risk period, the RI after the second dose was significantly larger than the RI after the first dosage in that risk period.

Two sensitivity analyses were performed. The first sensitivity analysis, as described above, sampled 80 percent of VTE reports randomly and assumed that these reports would eventually match the Brighton Collaboration requirements. We performed SCCS analysis again on the new dataset to check whether medical record reviews based on Brighton Collaboration criteria will affect the study results significantly. The second analysis investigated deceased cases’ impact on our results, as the original SCCS method assumed that events do not censor the remaining observation period [9,18].

We performed all statistical analysis using R software, version 4.0.3 (R Foundation for Statistical Computing, Vienna, Austria).

## 3. Results

### 3.1. Descriptive Analysis

We identified 2777 individuals who experienced VTE after receiving COVID-19 mRNA vaccination up to 20 January 2022 (Table 1). There was no gender preponderance discovered, and nearly half of the patients were over 60 (1315 cases, 47.3%). The median age was 59 years (interquartile range 45–70). Most cases (72.4% for the mRNA-1273 vaccine, 70.3% for the BNT162b2 vaccine) developed VTE within the indicated dose time interval, with the median time to VTE onset (days) being 10 (interquartile range 3–27) days. Cases with a VTE onset date that was later than the recommended dose time interval or that had an ambiguous VTE onset time were kept in the demographic and clinical analysis. These instances, however, were eliminated from the SCCS studies.

Half of the cases, 1413 (50.9%), were in severe condition, including being hospitalized, with a life-threatening condition, permanent disability, or death. Seventy-seven deaths were reported.

### 3.2. RI from SCCS Analysis

We found no safety problems after two doses of bothCOVID-19 mRNA vaccines (Table 2). That is to say, the risk of VTE in the three risk periods did not differ significantly from the reference period for each dose. On the other hand, the VTE risk did not change significantly for all of the risk periods between doses. Specifically, the risk of VTE following the second dose was not significantly higher than the risk following the first dose.

A subgroup analysis that separated VTE reports into female and male groups found no significant increase in any risk periods (Table 3). The majority of VTE cases were observed between 1 and 7 days after vaccination. However, SCCS analysis found that the risk of VTE did not alter appreciably in any risk period. On the other hand, the RI after the second dose did not increase significantly compared to the RI after the first dose. Even though the VTE risk on day zero after the second mRNA-1273 vaccination dosage was much lower (0.31, 95 percent CI, 0.12–0.79) than the first dose, the RI for each dose did not change significantly when compared to the reference period.

Based on the population median age of VTE reports, we also performed SCCS analysis in older (age ≥ 60) and younger (age < 60) age groups (Table 4). We do not find any significantly increased VTE risk among the pre-specified risk periods after COVID-19 mRNA vaccination.

We investigated whether the interaction between age and sex may affect the association. For each gender, we classified the reports into older (age ≥ 60) and younger (age < 60) groups. However, we did not find evidence that can relate the VTE onset with COVID-19 mRNA vaccination (Appendix A).

Two sensitivity analyses were involved in assessing the robustness of our results. The first sensitivity analysis found no safety signals (Appendix A). Therefore, our conclusion may still hold if medical assessment using the Brighton Collaboration Criteria of VTE is involved. The second sensitivity analysis did not find any significant association between VTE onset and COVID-19 mRNA vaccination (Appendix A), suggesting that censored cases may not affect our analysis results.

## 4. Discussion

Despite the many VTE cases identified, our real-world SCCS study in VAERS found no significantly elevated risk after the first or second dose of COVID-19 mRNA immunization. These conclusions may be held if the medical record review based on Brighton Collaboration criteria involves or excludes all deceased cases. Our finding agrees with a real-world evidence-based study in the US Mayo Clinic Health System [5]. Furthermore, a recent Danish retrospective cohort research demonstrated no statistically significant link between the onset of thrombotic or thrombocytopenic events after BNT162b2 immunization [19], indicating that our results may be applicable to other races. Our findings, however, contradict a disproportionality analysis performed on the World Health Organization’s global pharmacovigilance database (i.e., VigiBase) [4]. Therefore, further investigation is necessary to clarify the association between COVID-19 mRNA vaccination and VTE events.

Our research had some limitations. First, we addressed the underreporting problem in the VAERS dataset by assuming an exponentially decreasing reporting probability, which may be a strong assumption. However, a comprehensive simulation and real-world analysis revealed that the functional form of the reporting probability had little effect on the results [14]. Second, our study was not able to investigate the causal relationship between COVID-19 mRNA vaccination and VTE onset, which must be evaluated via a rigorous randomized clinical trial. Third, we excluded VTE reports with a time to VTE onset greater than the recommended dose interval for each vaccine. Although rare, these inaccuracies may contribute further bias to our estimation. Fourth, our analysis did not consider the underlying medical conditions, as such information is unavailable for some patients. However, the clinicians may consider the underlying medical conditions in the clinical review process if such information is available. Fifth, this study did not consider genetic mutation. A recent case report described a patient who developed a right leg deep venous thrombosis and pulmonary embolism one day after receiving the second dose of the BNT162b2 vaccine. Thrombophilia screening showed this patient was heterozygous for the FVL G169A mutation and homozygous for the MTHFR A1298C mutation [1]. This finding suggests that genetic mutation may be a potential risk factor. However, such genetic mutation information is rarely available in a pharmacovigilance database. Sixth, the identified VTE cases lacked rigorous clinical verification due to the poor clinical description in the VAERS. We tried to address the issue through sensitivity analysis, assuming that a subset of cases will ultimately be identified as VTE cases. Finally, this study was confined to mRNA vaccinations and the US population, with no control group to evaluate this association unbiasedly. This research might be expanded to include other populations or non-mRNA vaccines and to study the VTE incidence rate change after immunization in diverse groups.

## 5. Conclusions

We found no evidence of a potential VTE signal following COVID-19 mRNA immunization based on VAERS data. Our findings were limited to VAERS’ inherent limitations, including the lack of medical records. The COVID-19 vaccine remains the most effective tool for combating this pandemic. Any concerns about its potential AEs should be weighed against the advantages of vaccination.

## Figures and Tables

**Table 1 vaccines-10-01317-t001:** Demographics of venous thromboembolism reports after COVID-19 mRNA vaccination, as of 28 January 2022.

	Venous Thromboembolism (N = 2777)			
	mRNA-1273 (%)		BNT162b2 (%)	
**Characteristics**	After dose 1	After dose 2	After dose 1	After dose 2
**Reported sex**				
No.	670	680	567	860
Male	295 (44)	328 (48.2)	246 (43.4)	415 (48.3)
Female	367 (54.8)	351 (51.6)	316 (55.7)	439 (51)
Unknown	8 (1.2)	1 (0.1)	5 (0.9)	6 (0.7)
**Age, y**				
Median (IQR)	59 (46–70)	63 (47.75–72)	54 (40.75–66)	59 (45–69)
12–17	0 (0)	0 (0)	10 (1.8)	13 (1.5)
18–29	25 (3.7)	28 (4.1)	33 (5.8)	42 (4.9)
30–39	74 (11)	56 (8.2)	88 (15.5)	92 (10.7)
40–49	98 (14.6)	96 (14.1)	96 (16.9)	132 (15.3)
50–59	128 (19.1)	100 (14.7)	102 (18)	161 (18.7)
60–69	145 (21.6)	171 (25.1)	114 (20.1)	200 (23.3)
70–79	115 (17.2)	141 (20.7)	76 (13.4)	135 (15.7)
≥80	48 (7.2)	72 (10.6)	33 (5.8)	64 (7.4)
Unknown	37 (5.5)	16 (2.4)	15 (2.6)	21 (2.4)
**Time to VTE onset, d**				
Median (IQR)	9 (3–21)	12 (3–35)	6 (2–14)	13 (3–38.75)
≤28 d after mRNA-1273 vaccination	519 (77.5)	458 (67.4)	NA	NA
≤21 d after BNT162b2 vaccination	NA	NA	481 (84.8)	522 (60.7)
Unknown	39 (5.8)	13 (1.9)	20 (3.5)	22 (2.6)
**Seriousness ^a^**				
Hospitalized	334 (49.9)	354 (52.1)	274 (48.3)	450 (52.3)
Life-threatening condition	200 (29.9)	268 (39.4)	208 (36.7)	283 (32.9)
Permanent disability	28 (4.2)	42 (6.2)	33 (5.8)	64 (7.4)
Died	23 (3.4)	24 (3.5)	13 (2.3)	17 (2)
Unknown	272 (40.6)	229 (33.7)	226 (39.9)	319 (37.1)
**Recovered ^b^**				
Yes	170 (25.4)	161 (23.7)	122 (21.5)	183 (21.3)
No	289 (43.1)	323 (47.5)	317 (55.9)	478 (55.6)
Unknown	211 (31.5)	196 (28.8)	128 (22.6)	199 (23.1)

^a^ These conditions are determined by the reporter’s statement and are not mutually exclusive. ^b^ Patients’ recovery status at the time when the dataset was summarized.

**Table 2 vaccines-10-01317-t002:** Ratios of relative incidences and relative incidences of venous thromboembolism after two doses of COVID-19 mRNA vaccine.

		No. of Events ^a^		RI after Dose 1 (95% CI)	RI after Dose 2 (95% CI)	Ratio of RIs (95% CI) ^b^
		After Dose 1	After Dose 2			
**mRNA-1273**						
	No. of days after vaccination, d					
	0 ^c^	56	31	1.19 (0.65, 2.18)	0.81 (0.42, 1.55)	0.68 (0.41, 1.14)
	1–7	216	234	0.9 (0.56, 1.45)	1.2 (0.74, 1.95)	1.33 (0.96, 1.85)
	8–14	129	97	0.96 (0.68, 1.36)	0.89 (0.61, 1.29)	0.92 (0.63, 1.35)
	15–28	108	87	1 [Reference]	1 [Reference]	1 [Reference]
**BNT162b2**						
	No. of days after vaccination, d					
	0 ^c^	47	58	0.51 (0.26, 1.01)	0.62 (0.31, 1.21)	1.22 (0.73, 2.03)
	1–7	258	268	0.64 (0.38, 1.09)	0.66 (0.39, 1.12)	1.02 (0.7, 1.49)
	8–14	108	127	0.65 (0.45, 0.96)	0.76 (0.52, 1.1)	1.16 (0.76, 1.77)
	15–21	59	56	1 [Reference]	1 [Reference]	1 [Reference]

Abbreviations: CI, confidence interval; RI, relative incidence. ^a^ Of the 2777 venous thromboembolism reports, our analysis excluded (i) reports whose time to VTE onset was unknown or exceeded the recommended dose interval (28 days for the mRNA-1273 vaccine and 21 days for the BNT162b2 vaccine) to ensure reports were related to a single dose; (ii) reports with unknown dose series; (iii) reports with unknown age, younger than 12 years or 18 years when receiving the BNT162b2 vaccine or the mRNA-1273 vaccine, respectively; and (iv) duplicated reports. The final analysis included 1939 reports. ^b^ The ratios were estimated by dividing the relative incidences after dose 2 by the relative incidences after dose 1. ^c^ Zero refers to the day of vaccination.

**Table 3 vaccines-10-01317-t003:** The ratios of relative incidences and relative incidences of venous thromboembolism after two doses of COVID-19 vaccination, stratified by reported gender.

		No. of Events ^a^		RI after Dose 1 (95% CI)	RI after Dose 2 (95% CI)	Ratio of RIs (95% CI) ^b^
		After Dose 1	After Dose 2			
**Female**						
**mRNA-1273**						
	No. of days after vaccination, d					
	0 ^c^	33	24	1.25 (0.56, 2.82)	1.29 (0.55, 3.02)	1.03 (0.55, 1.96)
	1–7	113	111	0.83 (0.44, 1.58)	1.16 (0.60, 2.25)	1.39 (0.89, 2.18)
	8–14	61	54	0.78 (0.48, 1.25)	0.98 (0.59, 1.63)	1.26 (0.75, 2.10)
	15–28	65	42	1 [Reference]	1 [Reference]	1 [Reference]
**BNT162b2**						
	No. of days after vaccination, d					
	0 ^c^	31	40	0.66 (0.26, 1.66)	0.93 (0.37, 2.32)	1.41 (0.72, 2.74)
	1–7	138	141	0.67 (0.32, 1.39)	0.75 (0.36, 1.56)	1.11 (0.66, 1.89)
	8–14	61	67	0.71 (0.42, 1.20)	0.85 (0.51, 1.44)	1.20 (0.67, 2.15)
	15–21	30	27	1 [Reference]	1 [Reference]	1 [Reference]
**Male**						
**mRNA-1273**						
	No. of days after vaccination, d					
	0 ^c^	23	7	1.14 (0.46, 2.80)	0.35 (0.11, 1.02)	0.31 (0.12, 0.79)
	1–7	102	123	1.00 (0.50, 2.04)	1.24 (0.62, 2.51)	1.23 (0.76, 2.00)
	8–14	67	43	1.21 (0.72, 2.03)	0.79 (0.46, 1.37)	0.66 (0.38, 1.15)
	15–28	43	45	1 [Reference]	1 [Reference]	1 [Reference]
**BNT162b2**						
	No. of days after vaccination, d					
	0 ^c^	16	18	0.41 (0.14, 1.14)	0.38 (0.14, 1.05)	0.94 (0.41, 2.15)
	1–7	119	126	0.69 (0.31, 1.52)	0.61 (0.28, 1.32)	0.88 (0.51, 1.52)
	8–14	47	60	0.65 (0.37, 1.16)	0.69 (0.41, 1.19)	1.06 (0.57, 1.97)
	15–21	27	29	1 [Reference]	1 [Reference]	1 [Reference]

Abbreviations: CI, confidence interval; RI, relative incidence. ^a^ Twenty cases that reported unknown or missing sex were excluded. ^b^ The ratios were estimated by dividing the relative incidences after dose 2 by the relative incidences after dose 1. ^c^ Zero refers to the day of vaccination.

**Table 4 vaccines-10-01317-t004:** The ratios of relative incidences and relative incidences of venous thromboembolism after two doses of COVID-19 vaccination, stratified by reported age.

		No. of Events ^a^		RI after Dose 1 (95% CI)	RI after Dose 2 (95% CI)	Ratio of RIs (95% CI) ^b^
		After Dose 1	After Dose 2			
**Age < 60**						
**mRNA-1273**						
	No. of days after vaccination, d					
	0 ^c^	32	14	1.37 (0.58, 3.24)	1.15 (0.43, 3.01)	0.84 (0.39, 1.78)
	1–7	108	106	0.89 (0.45, 1.77)	1.67 (0.82, 3.46)	1.88 (1.15, 3.07)
	8–14	65	51	0.92 (0.56, 1.51)	1.38 (0.79, 2.43)	1.50 (0.86, 2.61)
	15–28	60	31	1 [Reference]	1 [Reference]	1 [Reference]
**BNT162b2**						
	No. of days after vaccination, d					
	0 ^c^	25	31	0.46 (0.19, 1.15)	0.83 (0.33, 2.06)	1.78 (0.89, 3.57)
	1–7	149	158	0.62 (0.31, 1.24)	0.94 (0.46, 1.94)	1.52 (0.92, 2.52)
	8–14	67	57	0.64 (0.39, 1.03)	0.78 (0.46, 1.33)	1.22 (0.69, 2.17)
	15–21	40	25	1 [Reference]	1 [Reference]	1 [Reference]
**Age ≥ 60**						
**mRNA-1273**						
	No. of days after vaccination, d					
	0 ^c^	18	17	0.68 (0.27, 1.67)	0.52 (0.21, 1.28)	0.77 (0.36, 1.65)
	1–7	103	125	0.81 (0.41, 1.61)	0.80 (0.41, 1.56)	0.99 (0.62, 1.57)
	8–14	62	45	0.97 (0.59, 1.62)	0.57 (0.34, 0.96)	0.59 (0.34, 1.01)
	15–28	45	54	1 [Reference]	1 [Reference]	1 [Reference]
**BNT162b2**						
	No. of days after vaccination, d					
	0 ^c^	22	26	0.52 (0.18, 1.52)	0.37 (0.13, 1.02)	0.70 (0.32, 1.53)
	1–7	101	107	0.59 (0.25, 1.39)	0.37 (0.16, 0.83)	0.63 (0.35, 1.14)
	8–14	40	69	0.65 (0.35, 1.24)	0.67 (0.39, 1.15)	1.03 (0.53, 1.98)
	15–21	19	31	1 [Reference]	1 [Reference]	1 [Reference]

Abbreviations: CI, confidence interval; RI, relative incidence. ^a^ Ten cases that reported unknown or missing age were excluded. ^b^ The ratios were estimated by dividing the relative incidences after dose 2 by the relative incidences after dose 1. ^c^ Zero refers to the day of vaccination.

## Data Availability

Publicly available datasets were analyzed in this study. The data can be found here: https://vaers.hhs.gov/data/datasets.html (accessed on 13 February 2022).

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
