# Peer review of "Venous Thromboembolism following Two Doses of COVID-19 mRNA Vaccines in the US Population, 2020–2022"

_vaccines, 2022, doi:10.3390/vaccines10081317_

Round 1

Reviewer 1 Report

1.     Abstract needs to be improved.

2.    “ safety signal is definitely”  This is vague word.

3.    Because clinical trials are known to have low power in detecting the signal of many infrequent AEs, post-marketing surveillance is essential in evaluating vaccination safety.” This is not true. This is one of the objective of clinical trial to find safety and any side effects.

4.    However, research on the VAERS dataset is frequently 39 hampered by data limitations, such as case underreporting and the lack of a control group 40 to assess excess vs. baseline risk in the population of interest” How authenticity of any false positive is figured.

5.    Informed 64 consent was waived as the VAERS dataset is fully anonymized.” What is anonymized ?

6.    A sensitivity analysis, which  that 80% of cases could ultimately satisfy the Brighton Collaboration criteria for 92 VTE, was performed to compensate for this limitation. ”  How sensitivity analysis criteria was reached.

7.    Introduction should provide more background on sccs analysis.

8.    USA has three major vaccine approved for COVID-19. Why only data from two major vaccines are reported. Is Johnson and Johnson vaccine has so such side effects? As an control did the data from any other vaccine or disease condition exist where authors look at Venous Thromboembolism.

9.    For the two vaccine people have received booster. Is there any change in Venous Thromboembolism after each dosage.

10. In Table 1 what is meaning of recovery ? Does it mean subject died ? If so why numbers don’t match with column above.

11. In Table 2 , after 95% interval column what is the bracket is not labeled.1

12. Is there any such data available from any other country, Europe or Asia where such SCCS analysis can be performed in Caucasian or any other ethnicity/race.

13. The study results will be much better represented by Kaplan Meier or any other appropriate graphical representation.

14.  A graph of study geographical map and time would be helpful.

15.  A flow diagram with inclusion and exclusion criteria will be helpful.

16.  Does the study follow PRISMA or any other specific guidelines?

Reviewer 2 Report

Based on the US Vaccine Adverse Event Reporting System (VAERS) data, this study investigated the association between COVID-19 mRNA vaccination and the risk of venous thromboembolism events among US adults. Based on existing divergence, although this research exists some limitations, I think this study is of great value. I recommend this research article for publication in Vaccines after minor revision. Some concerns and suggestions are listed as follows.

1. “COVID-19” is the abbreviation of “Coronavirus Disease 2019”, so the word “disease” after “COVID-19” at line 13 should be deleted. Line 12, I recommend to change the sentence “COVID-19 mRNA vaccine is probably the most effective strategy to eradicate the COVID-19 disease.” as “COVID-19 mRNA vaccine is one the most effective strategies to fight against the COVID-19.”

2. Line 20, since “COVID-19” has already been listed, “COVID-19” in the keyword “COVID-19 mRNA vaccines” should be removed.

3. A full stop is used at the end of a sentence, so please put the full stop “.” after the number of references in the main text.

4. Line 123-124, please change “half” as “nearly half” because 47.3% is not half. And the full stop “.” after 60 should be removed.

5. Line 171, please change “are” as “is”.

6. The format of references should be revised according to the requirement of “Vaccines”.

Reviewer 3 Report

The article describes a research about possible relation between the VTE and COVID-19 mRNA vaccination. The research was done, finalized and it should be presented in past tense, not in present tense. Also, some sentences are too complicated making it rather difficult to follow the text.

Reviewer 4 Report

In this paper, the author investigated the correlation between COVID-19 vaccination and a possible side effect--venous thromboembolism (VTE). Using the database from the US Vaccine Adverse Event Reporting System (VAERS), such correlation is not found, indicating that VTE event is not significantly associated with vaccination.

Overall, this research topic is important. The method and the datasets are properly described. The potential limitation of the analysis is also discussed.  

I feel like more reference should be cited.

Reviewer 5 Report

The article with title " Venous Thromboembolism following Two Doses of COVID-19 mRNA Vaccines in the US Population, 2020-2022a", authors by Daoyuan Lai and Coleagues provide a comprehensive result of VTE Adverse Events following vaccination with mRNA vaccines.

Major points

1.       As a limitation of this study, please describe the point that the underlying medical condition was not considered in the analysis.

2.       Can the authors discuss if the results of the association between VTE and mRNA vaccines are noticed similarly in other races (For example, Danish study, Hviid A et al., 2022, Ann Intern Med 175:541-546)

3.       Last major point to develop in the discussion: Authors do not take into account the genetic mutation, Atoui et al. describe a patient, heterozygous for the FVL G169A mutation and homozygous for the MTHFR A1298C mutation, who developed a right leg deep venous thrombosis and pulmonary embolism 24 h after a second dose of Pfzer BioNTech mRNA COVID vaccine.

Round 2

Reviewer 1 Report

The authors have answered some of my comments. The point on appropriate controls is not  answered since the comparison is only for mRNA vaccines and in US population. This reflects little bias from authors. There are sentences also “Second, it is well-known that COVID-19 mRNA vaccination will boost the chance of developing myocarditis”. There are some causal relation suggested but authors can not link the two based on evidence available or just by correlation. In my opinion authors have started to look for risk due to mRNA vaccines and keen on finding something or anything. In my humble opinion authors should be aware of confirmation bias and try to test there hypothesis in unbiased manner. The problem with this and any other study is lack of appropriate controls eg another vaccine or population based control. Also as with any therapeutic treatment the real analysis should be analysis of risk benefit ratio. The way this study is presented reads more like authors don’t believe in mRNA vaccines so we will associate some risk event associated with it. It is very important to study risk factor analysis of mRNA based vaccines in unbiased manner.

I suggest authors include a paragraph on limitataion of this study in clear manner.  

Reviewer 5 Report

I have no more comments.

Author Response

Thanks a lot for your comments and suggestions!

Round 3

Reviewer 1 Report

None

This manuscript is a resubmission of an earlier submission. The following is a list of the peer review reports and author responses from that submission.